# Infection with intestinal helminth (*Hymenolepis diminuta*) impacts exploratory behavior and cognitive processes in rats by changing the central level of neurotransmitters

Kamilla Blecharz-Klin[1], Magdalena Świerczyńska[1], Agnieszka Piechal[1], Adriana Wawer[1], Ilona Joniec-Maciejak[1], Justyna Pyrzanowska[1], Ewa Wojnar[1], Anna Zawistowska-Deniziak[2], Anna Sulima-Celińska[3], Daniel Młocicki[2,3]*, Dagmara Mirowska-Guzel[1]*

**1** Department of Experimental and Clinical Pharmacology, Medical University of Warsaw, Centre for Preclinical Research and Technology CePT, Warsaw, Poland, **2** W. Stefański Institute of Parasitology, Polish Academy of Sciences, Warsaw, Poland, **3** Department of General Biology and Parasitology, Medical University of Warsaw, Warsaw, Poland

* daniel.mlocicki@wum.edu.pl (DM); dagmara.mirowska-guzel@wum.edu.pl (DM-G)

## Abstract

Parasites may significantly affect the functioning of the host organism including immune response and gut-brain-axis ultimately leading to alteration of the host behavior. The impact of intestinal worms on the host central nervous system (CNS) remains unexplored. The aim of this study was to evaluate the effect of intestinal infection by the tapeworm *Hymenolepis diminuta* on behavior and functions of the CNS in rats. The 3 months old animals were infected, and the effects on anxiety, exploration, sensorimotor skills and learning processes were assessed at 18 months in Open Field (OF), Novel Object Recognition (NOR) and the Water Maze (WM) tests. After completing the behavioral studies, both infected and non-infected rats were sacrificed, and the collected tissues were subjected to biochemical analysis. The levels of neurotransmitters, their metabolites and amino acids in selected structures of the CNS were determined by HPLC. In addition, the gene expression profile of the pro- and anti-inflammatory cytokines (TNF-α, IL-1β, IL-6 and IL-10) was evaluated by Real-Time PCR to determine the immune response within the CNS to the tapeworm infection. The parasites caused significant changes in exploratory behavior, most notably, a reduction of velocity and total distance moved in the OF test; the infected rats exhibited decreased frequency in the central zone, which may indicate a higher level of anxiety. Additionally, parasite infestation improved spatial memory, assessed in the WM test, and recognition of new objects. These changes are related to the identified reduction in noradrenaline level in the CNS structures and less pronounced changes in striatal serotonergic neurotransmission. *H. diminuta* infestation was also found to cause a significant reduction of hippocampal expression of IL-6. Our results provide new data for further research on brain function during

**Data Availability Statement:** All relevant data is included in the figures and tables of the Manuscript.

**Funding:** This research was partially supported by the National-Science-Center-Poland (Grant Number 2014/13/B/NZ6/00881), awarded to DM. Project implemented with CePT infrastructure financed by the European Union - the European Regional Development Fund within the Operational Programme "Innovative economy" for 2007-2013". The funders had no role in study design, data collection and analysis, decision to publish, or preparation of the manuscript.

**Competing interests:** The authors have declared that no competing interests exist.

parasitic infections especially in relation to helminths and diseases in which noradrenergic system may play an important role.

## Author summary

Recent advances in the research on parasitic manipulation and/or control of the nervous system of their host resulted in the development of neuro-parasitology, a new and emerging branch of science. There have been advances in this area in relation to parasite-insect interactions or parasites directly invading central nervous system (CNS). However, the neuro-parasitology of parasitic infections in vertebrate hosts remains unexplored. In our study the effect of intestinal infection by the tapeworm on the behavior, neurotransmission and functions of the CNS in rats was evaluated. This infection positively influenced spatial memory and new object recognition. At the same time, the infected animals developed a greater level of anxiety and move more slowly. Behavioral changes were related to the reduction in noradrenaline level in the CNS structures, and less pronounced changes in striatal serotonergic neurotransmission. The results provide important data for the further progress in neuro-parasitology and our understanding of parasite-host interactions. In our opinion in the near future may turn out that the role of the intestinal host macrobiome in the CNS functioning may be just as significant as that of the microbiome. Presented neuro-immunological data provide a new perspectives for further studies on the CNS under intestinal parasite infection. The data of behavioral changes induced by active parasitic infection may be valid for explanations of the host-parasite relationship at the evolutionary level and their molecular adjustment.

## 1. Introduction

The small intestine is a hostile environment for parasitic worms including *Hymenolepis diminuta*, being subject to digestive enzymes, immune response components, bacteria and active peristaltic movements. To provide stability, adult parasites utilize their scolex and tegumental microtiches to anchor to the intestinal epithelium, thus allowing parasite-derived molecules like excretory-secretory, surface, and tegumental proteins to interact with the host immune system [1–2]. Many of these molecules are proteins involved in the parasite metabolism and survival strategies, and can play significant roles in the parasite-host relationship.

The life cycle of *H. diminuta* parasite involves rodents (especially rats) and incidentally humans as definitive hosts, and tenebrionid beetles (*Tribolium* spp. and *Tenebrio* spp.) as intermediate hosts. The eggs pass into the external environment along with the feces of the definitive host and, when eaten by the beetle, the hexacanth larva hatches and penetrate into the beetle haemocoel and undergoes transformation into a cysticercoid. This process takes about 10–14 days. When a rat or human swallow the infected insect, digestive enzymes in the stomach and duodenum cause activation of cysticercoid and its maturation into adult tapeworm, including production of proglottids with mature sexual organs. Tapeworms settle into the small intestine of the definitive hosts and mature sexually in 2–3 weeks [3].

The presence of adult tapeworms may influence the function of the host intestines, thereby affecting the intestinal processes, immunity and microbiome [4–6]. Studies into gut-brain crosstalk have revealed a complex communication system likely to have multiple effects on

behavior and cognitive functions [7]. The complexity of these interactions is held within the *gut-brain axis* (GBA).

Recent research has provided a new insight into the interactions taking place between enteric microbiota and the central and enteric nervous systems, indicating the causal relationship between gut microbiota and brain functions [8–10]. Therefore, it seems highly probable that also intestinal parasites, may influence their surrounding environment and, consequently, the brain function and behavior of the host.

Infective stimuli may activate some immune mechanisms which in turn causes remodeling of neural circuits, influences memory consolidation, long-term potentiation in hippocampus and neurogenesis. Immune-active molecules such as cytokines, may induce alterations in several neurotransmitters in the CNS but also may act themselves as immunotransmitters affecting neuronal function. Parasite invasion may have a serious effect on the host organism and may activate immune responses. These may result in phenotypic changes that could easily be interpreted as the effect of parasite manipulation [11]. One example meriting particular note is invasion of the host brain, which can alter the host behavior in many ways; however, the response of the brain to these infections remains unclear. In most cases, such invasions improve parasite dissemination by altering host behavior, what is predominantly considered as negative for the host. However, present study indicates that infection with the intestinal helminth *Hymenolepis diminuta* might have effects on the behavior of the definitive host and impact neurotransmission in the brain. It has been suggested that *H. diminuta* infection may offer therapeutic potential by protecting the host from autoimmune diseases [12]. Furthermore, experiments on animal models indicate that *H. diminuta* ameliorates inflammatory diseases such as inflammatory bowel disease, and protect against chemically-induced colitis [6,13]. Moreover, recent reports reveal that intestinal helminths, including *H. diminuta*, play important roles in modeling the composition of the intestinal microbiome and its impact on the functioning of the immune and nervous systems [14–16]. These studies suggest that early colonization by commensal organisms such as helminths causes modulation of the immune activity, thus allowing them to positively influence the development of neuroinflammatory and cognitive disorders. Williamson et al. [15,17] showed that neonatal colonization of male rats by helminths prevented inflammation-induced cognitive impairments. Pups inoculated with *H. diminuta* and then injected with live *E. coli* and LPS did not show any memory disturbances in a fear conditioning task. In addition, maternal helminth inoculation prevented the increase of hippocampal IL-1 mRNA in neonatally-infected pups.

Since only very limited available data focus on the effect of tapeworm infestation on behavior and brain function in a host, the aim of the present study was to evaluate the impact of *H. diminuta* tapeworm infection on rat behavior, with a particular emphasis on learning memory consolidation, anxiety, motor skills and the levels of selected neurotransmitters. It also examines the gene expression for pro- and anti-inflammatory cytokines in central nervous system (CNS) that may induce alterations in several neurotransmitters causing remodeling of neural circuits, may influence memory consolidation, long-term potentiation in hippocampus and neurogenesis. Immune-active molecules such as cytokines may act themselves as immunotransmitters affecting neuronal function and shapes the host behavior. New data on the impact of *H. diminuta* infection on cognitive processes and brain neurotransmission, and thus on the interactions between host and parasite was revealed. Our findings may serve as a valuable start for further research on the influence of parasitic infections on host brain function, and may indicate a new path in the study on neurodegenerative diseases characterized by noradrenergic system dysfunctions.

## 2. Results

### 2.1. Parasitic infection inhibited Open Field (OF) behavior

There were not significant differences in the rat body weight, physical condition (assessed by veterinarian) and mortality between the infected and uninfected rats.

It was found that in male rats, parasitic infection inhibited Open Field behavior. Compared to the control group, infected rats demonstrated a shorter total path length in the test (Con: 13.52±0.004 m, Hym: 11.084±0.003 m; $F_{(1,13)}$ = 11.5762, p = 0.004) (Fig 1A), lower velocity of movement (Con: 0.07±0.0002, Hym: 0.06±0.0002 m/s; $F_{(1,13)}$ = 12.435, p = 0.004) (Fig 1B) and smaller frequency central part of the apparatus (Con: 12.00±0.63, Hym: 7.00±0.54; $F_{(1,13)}$ = 17.843, p = 0.001) (Fig 1C) as well as spent less time in the central area (Con: 43.215±3.18 s, Hym: 26.726±3.38 s; $F_{(1,13)}$ = 6.316, p = 0.026) (Fig 1D). Aversion to the central zone may

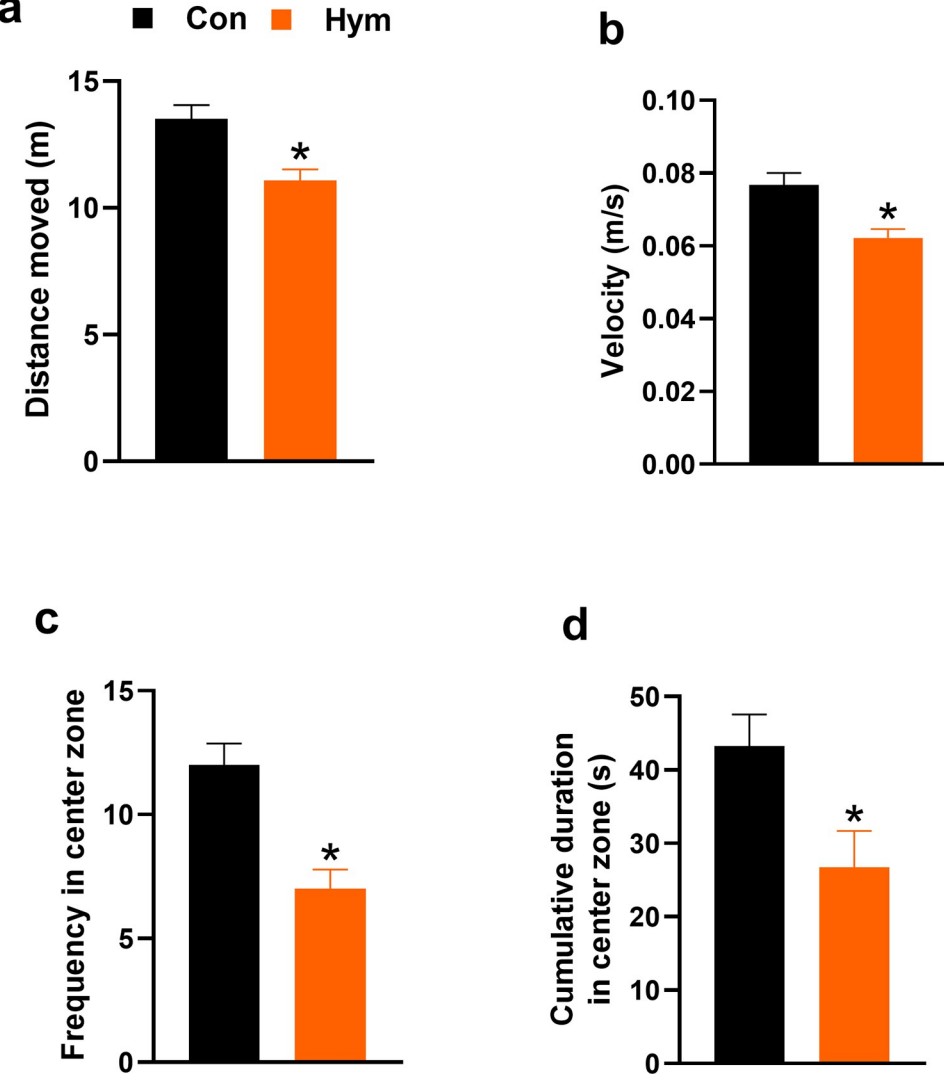

**Fig 1. Main Open Field parameters for the control rats (Con, n = 8) and rats infected with _H. diminuta_ (Hym, n = 7).** Infected animals exhibited reduced distance moved (a), velocity (b), frequency in the center zone (c) as well as reduced time spend in the central zone of the apparatus (d). * Hym _vs_ Con, p<0.05 (Newman-Keuls test). *** Hym _vs_ Con, p<0.005 (Newman-Keuls test).

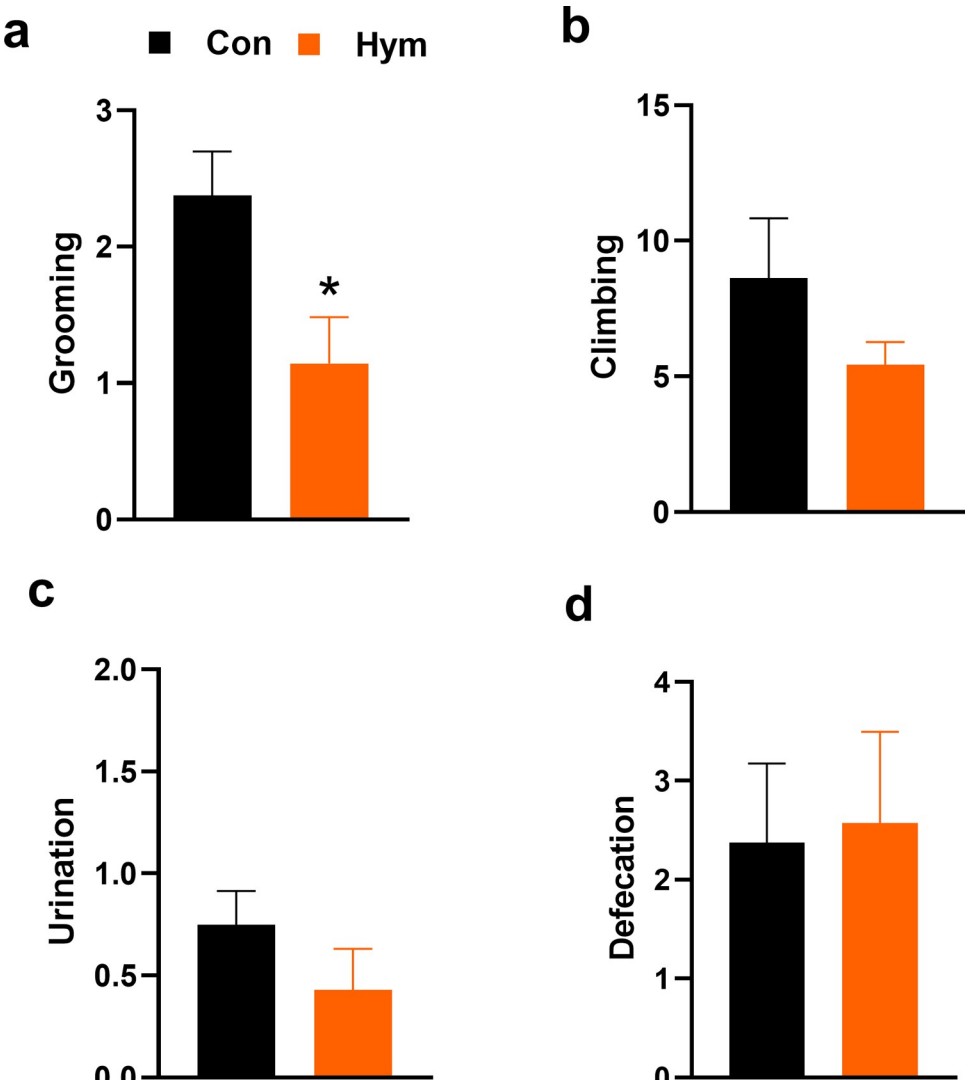

**Fig 2.** Total numbers of different exploratory behavior: grooming (a) and climbing (b) and the main determinants of stress: urination (c) and defecation (d) in the Open Field test for the control rats (Con, n = 8) and rats infected with *H. diminuta* (Hym, n = 7). * Hym *vs* Con, p<0.05 (Newman-Keuls test).

indicate a higher level of anxiety in infected rats. At the same time, other parameters that determine stress in animals such as number of urination (Con: 0.750±0.12, Hym: 0.429±0.14; $F_{(1,13)}$ = 1.560, p = 0.234) (Fig 2C) and defecation (Con: 2.375±0.58, Hym: 2.571±0.63; $F_{(1,13)}$ = 0.026, p = 0.874) (Fig 2D) did not change in infected animals compared to the control.

Infestation with *H. diminuta* also caused a reduction in grooming (Con: 2.375±0.24, Hym: 1.143±0.23; $F_{(1,13)}$ = 6.866, p = 0.021) (Fig 2A). However, the infected rats did not demonstrate any significant change in other exploratory parameters—climbing (Con: 8.625±1.61, Hym: 5.429±0.58; $F_{(1,13)}$ = 1.644, p = 0.222) (Fig 2B).

## 2.2. Novel Object Recognition (NOR) is higher in infected rats

During the familiarization phase (day 1) of NOR, the infected rats spent more total time on exploring two identical objects (Con: 10.165±0.887 s, Hym: 23.223±2.522 s; $F_{(1,13)}$ = 4.755,

p = 0.0482) but their velocity was lower (Hym: 0.043±0.006 m/s) compared to the control group (Con: 0.064±0.006 m/s) ($F_{(1,13)}$ = 6.228, p = 0.027).

*H. diminuta* infection had no effect on the other measured parameters, such as frequency of climbing on object A1 ($F_{(1,13)}$ = 1.053, p = 0.323) and A2 ($F_{(1,13)}$ = 0.043, p = 0.838), latency to first contact with any object ($F_{(1,13)}$ = 2.653, p = 0.127), cumulative time exploring object A1 ($F_{(1,13)}$ = 3.101, p = 0.102) and A2 ($F_{(1,13)}$ = 1.786, p = 0.204).

During the choice phase (day 2) when the animals had to choose between the familiar and the novel object infected animals climbed the new object (B) significantly more often than the control rats (Con: 0.625±0.32, Hym: 2.143±0.40; $F_{(1,13)}$ = 8.782, p = 0.011). However, no statistically significant differences were observed in the velocity ($F_{(1,13)}$ = 3.329, p = 0.091), total time of contact with both object (F = 2.741, p = 0.122), or the cumulative time exploring the old object ($F_{(1,13)}$ = 0.729, p = 0.409) or the new object ($F_{(1,13)}$ = 0.800, p = 0.387).

No differences in the value of Discrimination Index (DI) were found between experimental groups for the choice phase (Con: -0.292±1.212, Hym: -0.358±0.981; $F_{(1,13)}$ = 0.002, p = 0.968). A negative DI score indicated that the animals spent more time with a familiar object.

Infestation with *H. diminuta* did not significantly influence the Index of Global Habituation (GHI) determined, by comparing the total time spent in exploring the two objects during the familiarization phase to that spent in the test phase (Con: -0.684±0.350, Hym: -0.738±0.158; $F_{(1,13)}$ = 0.176, p = 0.897). A higher GHI usually indicates less interest in the objects in the next trial.

Recognition Index (RI)—the time spent investigating the novel object relative to the total object investigation—reflects good recognition memory. Parasite infection does not affect this parameter (Con: 2.294±1.039, Hym: 1.578±0.725; $F_{(1,13)}$ = 0.301, p = 0.593).

Summing up, it can be stated that the infected rats showed a slightly greater interest in the presented objects, which was expressed by a longer time of exploration objects during the familiarization phase and more frequent climbing on a new object during the choice phase. Similarly to the Open Field test, a reduced speed of movement of infected animals were also observed, but only on the first day of the test.

## 2.3. Water maze (WM) results

**2.3.1. Acquisition phase (days 1–4, trials 1–16).** During acquisition phase spatial learning is assessed across repeated trials where animals learn relies on distal cues to navigate from start locations to locate a submerged escape platform. Escape latency was defined as the time to find the platform during each trial and was used as an indicator of spatial learning. No statistically significant differences in the latency associated with learning the platform location (trials 1–16) were found between the control group (Con: 30.227±7.866 s) and infected rats (Hym: 29.357±8.845 s) during the four day trial, as evaluated by ANOVA with repeated measurements ($F_{(1,13)}$ = 0.036, p = 0.853).

The escape latency—ANOVA results did not show a significant mean effect for particular days: (day 1: Con: 43.59±7.862 s, Hym: 45.18±8.011 s, $F_{(1,58)}$ = 0.079, p = 0.780, day 2: Con: 30.94±7.428 s, Hym: 28.46±8.419 s, $F_{(1,58)}$ = 0.195, p = 0.660, day 3: Con: 20.69±6.364 s, Hym: 23.57±8.619 s $F_{(1,58)}$ = 0.298, p = 0.587, day 4: Con: 25.69±7.650 s, Hym: 20.21±7.464 s, $F_{(1,58)}$ = 1.034, p = 0.313) (Fig 3).

Significant differences were, however, found between groups in the velocity of swimming on days 3 and 4 of learning ($F_{(1,13)}$ = 9.488, p = 0.003 and $F_{(1,13)}$ = 11.611, p = 0.001 respectively). The rats infected with *H. diminuta* moved more slowly compared to the control animals (day 3: Con: 0.221±0.002 m/s, Hym: 0.188±0.001 m/s; day 4: Con: 0.223±0.001 m/s, Hym: 0.187±0.002 m/s). No differences in the speed of movement were noted on the remaining days, or in the distance travelled by the animals during the acquisition phase. The swim

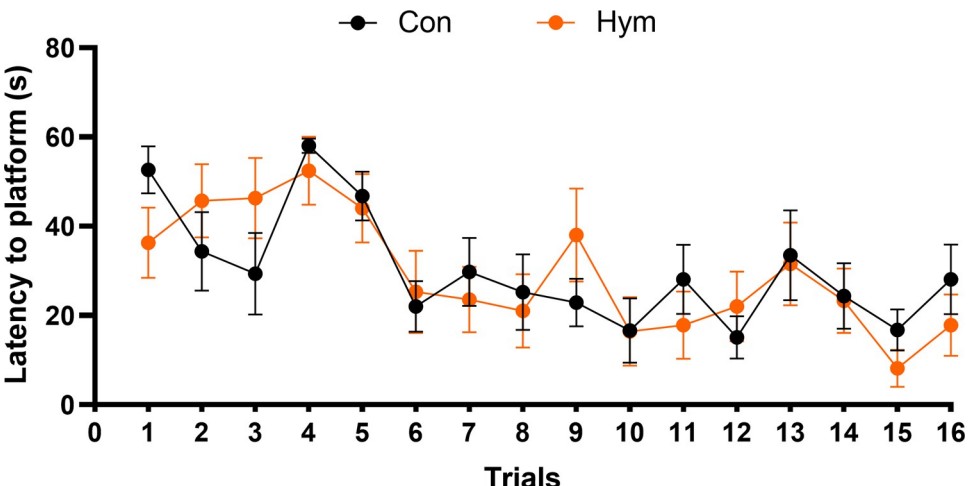

**Fig 3. Spatial learning (day 1–4, trial 1–16) in the control rats (Con, n = 8) and rats infected with *H. diminuta* (Hym, n = 7) in the Water Maze test.** Latency to the platform is parameter used to specify time to escape from water during acquisition trials using a submerged platform.

distance—ANOVA analysis for a particular day of training is presented as follows: (day 1: $F_{(1, 58)} = 0.001$, p = 0.977, day 2: $F_{(1, 58)} = 0.608$, p = 0.439, day 3: $F_{(1, 58)} = 0.045$, p = 8.833, day 4: $F_{(1, 58)} = 1.990$, p = 0.164). The total swim distance was the same among the groups (Con:6.496 ±0.65 m; Hym: 5.664±0.83m), ($F_{(1, 58)} = 0.769$, p = 0.384).

**2.3.2. Probe trial—memory test I (day 5, trial 17).** Reference memory is determined by preference for the platform area when the platform is absent during probe trial at the end of learning. The rats infected with *H. diminuta* demonstrated an insignificant increase in the number of crossings over the previous position of the platform during the memory test conducted on day 5 of the Water Maze (Con: 2.25±0.618, Hym: 3.29±0.746; $F_{(1,13)} = 1.159$, p = 0.301) (Fig 4A). A more detailed analysis showed that infected animals spent more time in the close zone around the platform (Con: 1.065±0.312 s, Hym: 2.280±0.426 s; $F_{(1,13)} = 5.462$, p = 0.036) as well as in the SE quadrant (Con: 15.225±1.164 s, Hym: 23.211±2.355 s; $F_{(1,13)} = 9.972$, p = 0.007). One-way ANOVA found no difference between the experimental groups, either in distance moved (Con: 1.31±0.054 m, Hym: 1.22±0.042 m; $F_{(1,13)} = 1.772$, p = 0.206) or velocity (Con: 0.22±0.009 m/s, Hym: 0.21±0.007 m/s; $F_{(1,13)} = 1.772$, p = 0.206).

**2.3.3. Repeated training (day 8, trials 18–21) indicates shorter latency to the hidden underwater platform in *H. diminuta*-infected rats.** After a two-day break, training resumed. Repeated training is necessary to remind the animal the position of the underwater platform. It was found that the infected group demonstrated significantly shorter latency to the hidden underwater platform (12.607±1.006 s) than the control group (20.625±0.880 s) ($F_{(1,13)} = 4.820$, p = 0.047). No significant difference in swimming distance ($F_{(1,13)} = 0.002$, p = 0.968) or speed ($F_{(1,13)} = 0.049$, p = 0.829) was found on the same day.

**2.3.4. Reversal platform (day 9, trials 22–25)–infected rats more frequently crossed the position of the old platform.** On day 9 of the Water maze test, the hidden platform was translocated to the NW quadrant. Reversal learning reveals if rats can extinguish initial learning of the platform's location and acquire a direct path to the new goal position.

ANOVA for repeated measures showed significant differences in frequency ($F_{(1,13)} = 4.842$, p = 0.032) and cumulative duration around old platform position ($F_{(1,13)} = 10.892$, p = 0.002). The rats with experimental hymenolepidosis crossed the position of the old platform more frequently then controls (Con: 0.219±0.173, Hym: 0.357±0.256) and spent more time around the

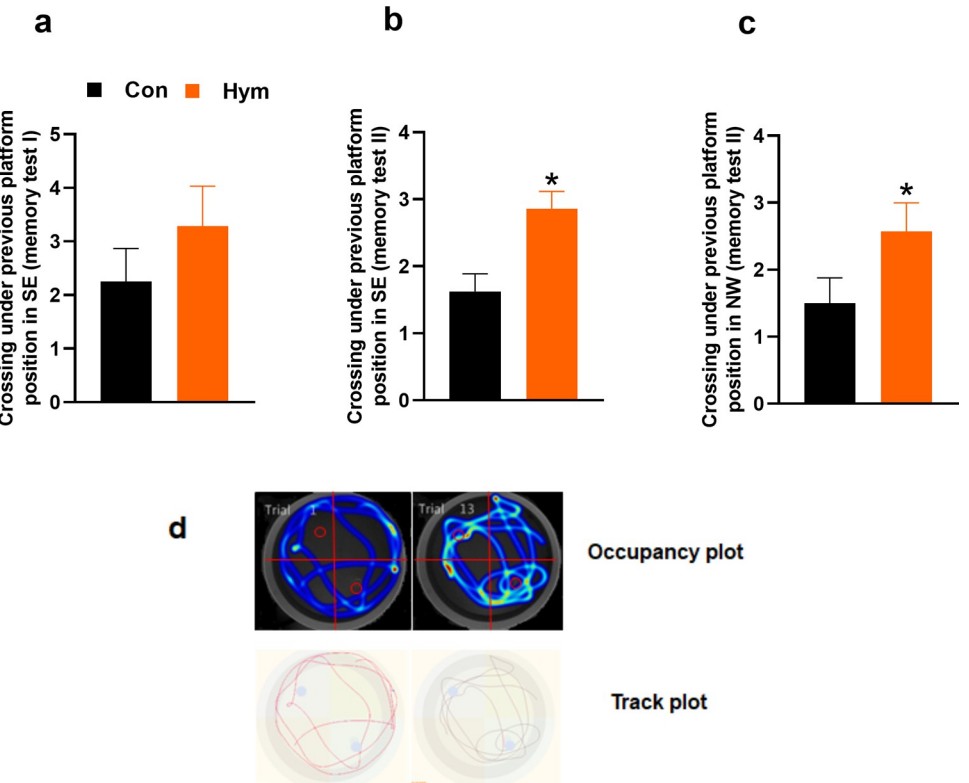

**Fig 4. Main results from the first and second probe trial of Water Maze test.** Effect of parasite infection on crossings under previous platform position in SE quadrant on first memory test (a) and crossings under previous (b) and new platform position in NW quadrant (c) during second memory probe trial as well as representative swim trajectory—track plot and heat maps–occupancy plot (d) of control rats (Con, n = 8) and rats infected with *H. diminuta* (Hym, n = 7) obtained by video-tracking software. Tracking analysis show detailed swim trajectory to determine the number of directed swim movements toward the previous platform position. Heat maps represent weighted occupancy across the entire 60s trial. Hot colors indicate longer dwell times. The platform area in target quadrants SE (old position of platform) and NW (new position of platform) are denoted with a circle. Each column represents the mean ± standard error of mean (SEM). ** Hym *vs* Con, p<0.01 (Newman-Keuls test).

previous position of the underwater platform (Con: 0.068±0.054, Hym: 0.120±0.097). No statistically significant differences were noted in latency to the new platform ($F_{(1,13)}$ = 0.001, p = 0.981) or cumulative time spent around the new platform location ($F_{(1,13)}$ = 0.002, p = 0.967). The swimming velocity was lower in the infected group than the control group ($F_{(1,13)}$ = 8.243, p = 0.006); however, no significant differences in swimming distance were observed between groups ($F_{(1,13)}$ = 0.227, p = 0.635).

**2.3.5. Reversal probe trial—memory test II (day 10, trial 26).** As is the case with the acquisition phase, also at the end of the reversal phase, a reversal probe trial is given 24 h later. The memory test on day 10 revealed significant differences in the number of crossings over the original position of the platform in the SE quadrant between the groups (Con: 1.625 ±0.263, Hym: 2.857±0.260; $F_{(1,13)}$ = 10.945, p = 0.006) (Fig 4B). Infected rats exhibited better memory of old platform position and tracking analysis: swim trajectory and heat maps that represent weighted occupancy across the entire 60s trial show that infected rats preferentially floated at previous location of platform in SE quadrant (Fig 4D). However, no difference was found between the treatment groups in the number of crossings over the new position of the platform in the NW sector according to one-way ANOVA (Con: 1.500±0.378, Hym: 2.571 ±0.428; $F_{(1,13)}$ = 3.545, p = 0.08) (Fig 4C).

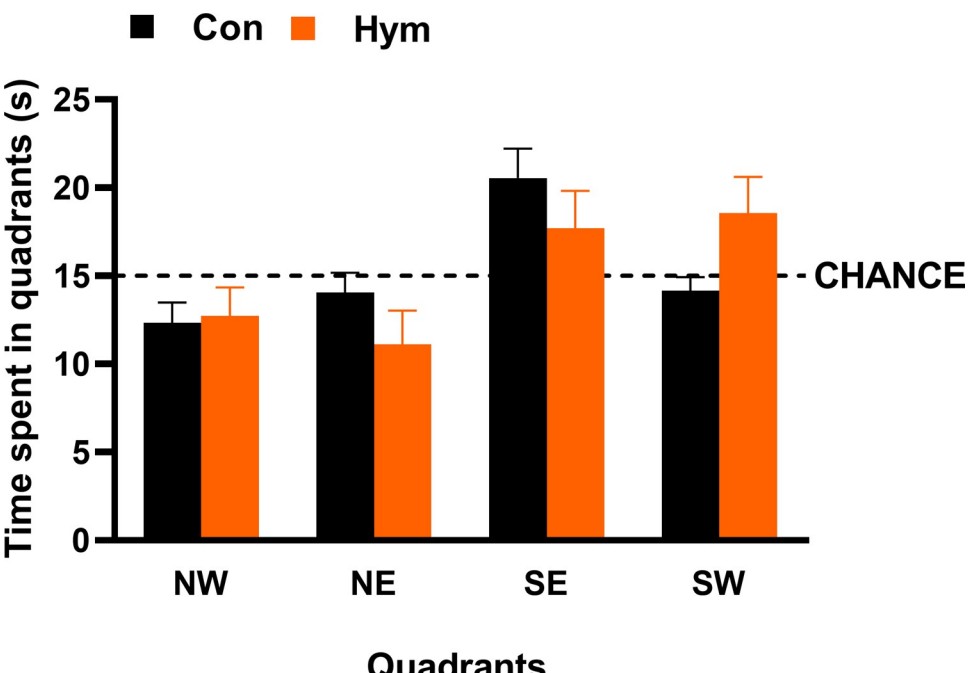

**Fig 5. Time spent in each four quadrants during the second memory probe trial of water maze (day 10) of control rats (Con, n = 8) and rats infected with *H. diminuta* (Hym, n = 7).** The horizontal dashed line marks chance performance.

Time spent in each four quadrants during the second memory probe trial of water maze was similar in both experimental groups (Fig 5).

No significant difference was found between the experimental groups for distance moved (Con: 1.43±0.052 m, Hym: 1.31±0.049 m; $F_{(1,13)} = 2.471$, p = 0.140) or velocity (Con: 0.24 ±0.009 m/s, Hym: 0.22±0.008 m/s; $F_{(1,13)} = 2.471$, p = 0.140).

**2.3.6. Visible platform test–cued test (day 11, trials 27–30).** On day 11, the dark-marked platform was raised above the water surface. The cued test checks the animals' ability to learn to swim to a cued goal. If animals are impaired in cued learning, there is a potentially serious concern about spatial learning capacity because cued learning requires the same basic abilities e.g. motoric ability, intact eyesight, and basic strategies. In this experiment no differences were found between the groups reading latency to reach the visible platform ($F_{(1,13)} = 0.099$, p = 0.961), distance moved ($F_{(1,13)} = 0.596$, p = 0.622) or velocity ($F_{(1,13)} = 1.511$, p = 0.227).

## 2.4. HPLC indicates alteration in neurotransmitters of infected rats

To obtain more detailed picture of the biochemical changes accompanying the parasitic infection in this experiment, we studied neurotransmitters and their metabolites in different CNS structures, especially those involved in learning processes (hippocampus, prefrontal cortex), but also those responsible for motor activity (cerebellum and medulla oblongata, striatum) and the hormonal status (hypothalamus). The infected rats demonstrated significantly altered levels of several neurotransmitters in distinct brain regions compared with uninfected animals (Table 1). The experimental hymenolepidosis in the rats was associated with a significant reduction of noradrenaline (NA) in the prefrontal cortex (Con: 148.34±21.85 pg/mg, Hym: 123.19±11.70 pg/mg; p = 0.032) and the cerebellum (Con: 195.37±10.12 pg/mg, Hym: 154.74 ±9.81 pg/mg; p = 0.018), as well as a decrease in the concentration of hippocampal 3-methoxy-

**Table 1. Monoamine and metabolite levels in the CNS structures of control rats (Con, n = 8) and rats infected with *H. diminuta* (Hym, n = 7) determined by high-performance liquid chromatography.**

| | Monoamine and metabolite levels in the CNS structures pg/g tissue (mean ± SEM) | | | | | | | | | | | |
|---|---|---|---|---|---|---|---|---|---|---|---|---|
| | hippocampus | | prefrontal cortex | | striatum | | hypothalamus | | cerebellum | | medulla oblongata | |
| | Con | Hym | Con | Hym | Con | Hym | Con | Hym | Con | Hym | Con | Hym |
| **NA** | 215.64 ±33.06 | 164.04 ±25.53 | **148.34 ±21.85** | **123.19 ±11.70*** | 255.99 ±14.45 | 49.78±11.87 | 1221.19 ±126.28 | 916.93 ±208.60 | **195.37 ±10.12** | **154.74 ±9.81*** | 302.46 ±26.11 | 310.39 ±37.57 |
| MHPG | **18.91 ±2.41** | **10.84 ±2.26*** | 29.34 ±4.72 | 21.87±4.22 | 17.13±3.57 | 22.34±3.95 | 1.98±0.87 | 1.41±1.09 | 1.32±0.40 | 0.53±0.22 | 1.04±0.28 | 0.88±0.44 |
| MHPG/NA | 0.11 ±0.03 | 0.09±0.03 | 0.18±0.03 | 0.19±0.05 | **0.23±0.11** | **0.57±0.16*** | n.d. | n.d. | n.d. | n.d. | n.d. | n.d. |
| **DA** | 800.36 ±65.11 | 872.31 ±130.09 | 427.66 ±39.56 | 428.31 ±24.87 | 892.33 ±97.25 | 759.67 ±39.13 | 594.01 ±31.82 | 538.19 ±103.11 | 37.94 ±1.70 | 38.41 ±4.42 | 394.76 ±14.35 | 397.98 ±31.59 |
| **DOPAC** | 206.59 ±52.76 | 250.26 ±91.70 | 252.54 ±34.42 | 227.91 ±36.37 | 1927.23 ±394.88 | 2824.79 ±501.73 | 536.15 ±54.85 | 536.32 ±32.77 | 24.67 ±2.11 | 22.75 ±2.22 | 35.01 ±4.39 | 36.96 ±5.13 |
| **DOPAC/ DA** | 0.30 ±0.09 | 0.42±0.20 | 0.63±0.12 | 0.53±0.08 | 2.36±0.57 | 3.77±0.73 | 0.92±0.10 | 7.50±6.58 | 0.66±0.07 | 0.63±0.10 | 0.09±0.01 | 0.10±0.01 |
| HVA | 15.01 ±7.85 | 50.90 ±25.98 | n.d. | n.d. | 454.45 ±91.53 | 664.13 ±116.06 | 48.16±6.55 | 65.63 ±15.08 | 35.02 ±2.14 | 30.40 ±2.98 | **42.49 ±3.17** | **25.33 ±3.85*** |
| **HVA/DA** | 0.02 ±0.01 | 0.09±0.05 | n.d. | n.d. | 0.56±0.13 | 0.87±0.16 | 0.08±0.01 | 1.85±1.77 | 0.94±0.08 | 0.87±0.16 | **0.11 ±0.01** | **0.06 ±0.01**** |
| **5-HT** | 298.09 ±53.41 | 319.06 ±48.72 | 114.25 ±37.81 | 163.95 ±31.59 | **469.76 ±69.32** | **231.06 ±14.08**** | 797.04 ±63.52 | 619.86 ±94.52 | 44.23 ±2.65 | 47.83 ±2.96 | 487.25 ±26.08 | 527.65 ±36.09 |
| **5-HIAA** | 158.52 ±122.26 | 250.99 ±20.40 | n.d. | n.d. | 4128.69 ±977.71 | 3401.25 ±632.38 | 258.27 ±37.51 | 178.66 ±26.40 | 9.23±1.13 | 5.91±0.82 | 45.69 ±1.75 | 47.41 ±3.00 |
| **5-HIAA/ 5-HT** | 0.69 ±0.48 | 1.25±0.95 | n.d. | n.d. | 10.55±3.17 | 15.55±3.41 | 0.32±0.04 | 0.30±0.03 | 0.21±0.02 | 0.13±0.02 | 0.09±0.00 | 0.09±0.01 |
| **3-MT** | n.d. | n.d. | n.d. | n.d. | 306.56 ±70.02 | 283.62 ±51.04 | 24.13±8.38 | 18.89±7.04 | 2.60±0.68 | 7.38±3.28 | 22.67 ±9.57 | 26.76 ±6.64 |

* Hym *vs* Control, p<0.05 (Mann-Whitney-U test)

** Hym *vs* Con, p<0.01 (Mann-Whitney-U test)

4-hydroxyphenylglycol (MHPG)—the main metabolite of noradrenaline (Con: 18.91±2.41 pg/mg, Hym: 10.84±2.26 pg/mg; p = 0.024). The decreased level of NA may be responsible for the decreased motility of infected animals observed in the Open Field and on the first day of the NOR. They also demonstrated an increase in MHPG/NA turnover calculated as the ratio of metabolite *vs* NA (Con: 0.23±0.11, Hym: 0.57±0.16; p = 0.013) and a reduced level of serotonin in the striatum compared to the control group (Con: 469.76±69.32 pg/mg, Hym: 231.06±14.08 pg/mg; p = 0.001). Similar changes but to a lesser extent concerned the dopaminergic system. The infected group also demonstrated a reduction of homovanillic acid (HVA) level in the medulla oblongata (Con: 42.49±3.17 pg/mg, Hym: 25.33±3.85 pg/mg; p = 0.018) and decrease in HVA/DA turnover (Con: 0.108±0.008, Hym: 0.063±0.009; p = 0.003).

The infected rats also displayed an increase in glutamic acid concentration in the striatum (Con: 893.16±66.22 ng/mg, Hym: 1095.52±23.76 ng/mg; p = 0.043) was observed (Table 2).

## 2.5. PCR results–a decrease in mRNA expression of hippocampal IL-6

The mRNA expression of the important proinflammatory factors in the brain structures was also determined using Real Time PCR. No significant differences were observed between groups for most tested factors; however, the rats infected with *H. diminuta* demonstrated decreased mRNA expression of hippocampal IL-6 (Con: 1.08±0.14, Hym: 0.63±0.05; $F_{(1,13)}$ = 8.426, p = 0.043) (Table 3). Gene expression levels represent the mRNA expression levels

**Table 2. Aminoacid levels in the CNS structures of control rats (Con, n = 8) and rats infected with *H. diminuta* (Hym, n = 7) determined by high-performance liquid chromatography.**

| | Aminoacids level in the CNS structures ng/mg tissue (mean ± SEM) | | | | | | | | | | | |
|---|---|---|---|---|---|---|---|---|---|---|---|---|
| | hippocampus | | prefrontal cortex | | striatum | | hypothalamus | | cerebellum | | medulla oblongata | |
| | Con | Hym | Con | Hym | Con | Hym | Con | Hym | Con | Hym | Con | Hym |
| **Glutamic acid** | 772.84 ±71.56 | 666.26 ±55.35 | 1151.39 ±63.82 | 1118.50 ±51.31 | **893.16 ±66.22** | **1095.52 ±23.76***  | 938.87 ±68.80 | 898.55 ±62.10 | 847.61 ±105.00 | 903.51 ±156.78 | 602.52 ±18.54 | 643.38 ±31.14 |
| **Taurine** | 538.75 ±72.55 | 470.0 3 ±83.91 | 1100.39 ±91.52 | 999.51 ±52.92 | 1417.30 ±136.37 | 1450.37 ±69.79 | 480.00 ±31.00 | 544.74 ±135.29 | 552.25 ±67.19 | 519.70 ±97.25 | 262.47 ±5.63 | 261.39 ±6.93 |
| **Alanine** | 56.48 ±2.62 | 56.27 ±4.78 | 115.01 ±4.42 | 106.72 ±5.23 | 87.28±5.41 | 106.26 ±7.44 | 44.86 ±2.40 | 109.37 ±63.26 | 41.53 ±5.25 | 42.45 ±7.90 | 52.61 ±2.88 | 52.55 ±2.91 |
| **γ-Aminobutyric acid** | 662.77 ±67.52 | 603.37 ±33.10 | 782.79 ±24.28 | 736.27 ±22.93 | 1198.92 ±165.53 | 835.02 ±48.99 | 904.14 ±62.21 | 721.52 ±133.25 | 225.24 ±28.63 | 231.91 ±43.39 | 238.60 ±12.28 | 231.67 ±12.22 |
| **Aspartic acid** | 374.50 ±18.85 | 329.13 ±25.47 | 550.94 ±22.52 | 538.21 ±62.24 | 392. 93 ±27.80 | 373.30 ±34.41 | 315.72 ±19.30 | 294.96 ±33.67 | 167.29 ±23.38 | 145.52 ±29.94 | 277.22 ±16.41 | 249.32 ±12.84 |
| **Histidine** | 15.06 ±0.55 | 12.91 ±0.66 | 22.21 ±1.46 | 19.88 ±1.46 | 20.68±0.91 | 22.78±2.87 | 15.91 ±1.14 | 19.32 ±3.92 | 10.15 ±1.22 | 10.13 ±1.73 | 9.31 ±0.45 | 9.09 ±0.39 |
| **Serine** | 82.45 ±8.39 | 76.75 ±10.38 | 120.31 ±7.41 | 116.83 ±6.41 | 132.11 ±4.16 | 156.58 ±18.56 | 57.73 ±3.35 | 59.55 ±5.77 | 44.43 ±5.46 | 43.94 ±7.35 | 46.86 ±1.62 | 44.75 ±0.97 |
| **Glutamine** | 698.13 ±40.67 | 642.38 ±59.33 | 905.43 ±56.57 | 937.99 ±50.91 | 966.04 ±50.69 | 1007.98 ±47.65 | 905.70 ±54.57 | 821.25 ±67.41 | 699.70 ±90.71 | 739.53 ±135.04 | 553.82 ±12.19 | 577.81 ±25.19 |
| **Arginine** | 133.16 ±9.88 | 150.56 ±14.02 | 106.52 ±4.36 | 112.64 ±6.7 | 236.02 ±14.38 | 218.93 ±11.42 | 65.77 ±5.87 | 74.31 ±11.97 | 39.13 ±7.39 | 36.46 ±8.31 | 77.20 ±6.31 | 92.38 ±7.59 |
| **Threonine** | 73.29 ±7.00 | 87.79 ±3.68 | 58.78 ±3.92 | 53.16 ±3.39 | 65.10±2.44 | 63.60±3.98 | 53.19 ±3.05 | 52.55 ±7.23 | 45.39 ±4.93 | 42.41 ±7.03 | 45.18 ±2.11 | 40.32 ±2.57 |

* Hym *vs* Control, $p < 0.05$ (Mann-Whitney-U test)

relative to the control. Although most of results are not statistically significant, continuous negative trends are evident for most measured cytokines in the hippocampus (Fig 6).

## 3. Discussion

The present study uses a rat model to analyze the impact of infection with *Hymenolepis diminuta* on behavioral parameters and neurotransmission in the host brain. The findings confirm the presence of a parasite-host interrelationship resulting in the modification of noradrenergic and serotonergic neurotransmission, manifested by changes in behavior and impaired locomotor activity in the infected animals. More specifically, the rats infected with *H. diminuta* exhibited inferior exploratory behavior and motor skills accompanied by increased

**Table 3. Real-time quantitative RT-PCR analyses for control rats (Con, n = 8) and rats infected with *H. diminuta* (Hym, n = 7).** Gene expression levels represent the mRNA expression for IL-6, IL-10, IL-1β and TNF-α in the hippocampus, prefrontal cortex, striatum and the cerebellum levels relative to the control. Values represent the means ± SEM for two experimental groups.

| | Relative expression (± SEM) | | | | | | | |
|---|---|---|---|---|---|---|---|---|
| | Hipocampus | | Prefrontal cortex | | Striatum | | Cerebellum | |
| Target gene | Con | Hym | Con | Hym | Con | Hym | Con | Hym |
| IL-6 | **1.08 ± 0.14** | **0.63 ± 0.05*** | 1.10 ± 0.18 | 1.37 ± 0.31 | 1.04 ± 0.10 | 1.50 ± 0.30 | 1.05 ± 0.35 | 1.20 ± 0.17 |
| IL-10 | 1.28 ± 0.29 | 0.65 ± 0.34 | 1.11 ± 0.20 | 1.57 ± 0.19 | 1.09 ± 0.18 | 2.33 ± 1.25 | 1.11 ± 0.53 | 1.48 ± 0.35 |
| IL-1β | 1.18 ± 0.31 | 0.91 ± 0.12 | 1.04 ± 0.13 | 1.03 ± 0.11 | 1.05 ± 0.13 | 1.53 ± 0.40 | 1.07 ± 0.14 | 1.13 ± 0.19 |
| TNF-α | 1.08 ± 0.14 | 0.87 ± 0.22 | 1.15 ± 0.23 | 1.10 ± 0.15 | 1.01 ± 0.05 | 1.41 ± 0.22 | 1.05 ± 0.12 | 0.96 ± 0.18 |

* Hym *vs* Control, $p < 0.05$ (Newman-Keuls test)

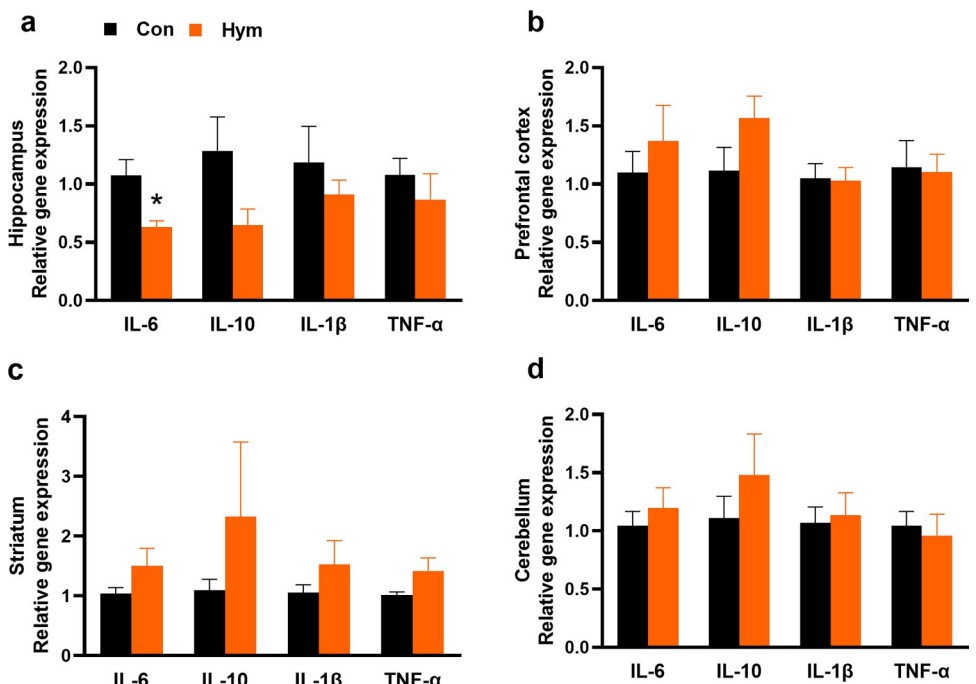

**Fig 6. Real-time quantitative RT-PCR analyses for control rats (Con, n = 8) and rats infected with *H. diminuta*** **(Hym, n = 7).** Gene expression levels represent the mRNA expression for IL-6, IL-10, IL-1β and TNF-α in the hippocampus (a), prefrontal cortex (b), striatum (c) and the cerebellum (d) relative to the control. Values represent the means ± SEM for two experimental groups. * Hym *vs* Control, p<0.05 (Newman-Keuls test).

thigmotaxis (the tendency to remain close to vertical surfaces) in the Open Field test. This slower movement speed may be caused by a decrease in noradrenalin level in the prefrontal cortex, striatum, and cerebellum. The decreased number of animals presences in the central zone of the apparatus may indicate a higher level of anxiety in the infected rats but it is necessary to confirm this effect using other behavioral stressful tests and/or by measuring the secretion of stress hormones. From an evolutionary perspective, anxiety has adaptive significance, allowing a creature to respond to environmental changes, and brings some advantages such as improved danger recognition; when they are functioning properly, these mental states can modulate behavior and increase sensitivity to signs of potential danger. Anxiety is also a component of de-escalating strategies aimed at maintaining social homeostasis [18]. Infection was also found to decrease other important elements of animal activity, such as grooming; self-grooming is a maintenance behavior that provides the animal physiological status, comfort and appearance. Striatal MHPG/NA turnover that reflects noradrenaline metabolism was elevated in infected animals. This may indicate higher level of stress in response to exposing to the experimental conditions. Simultaneously it cannot be excluded that some behavioral deficits could by associated with malaise, fatigue or nutritional deficits associated with *H. diminuta* infection and may not related strictly to anxiety or an altered emotionality. It should be taken into account that the parasite burden and the age of the infected rats may also play a role. Nevertheless, our research indicated differences between infected and non-infected animals pointing interesting direction for further research.

Contrary to previously published observations [19], our findings indicate that parasite infestation improved spatial memory, assessed in the Water Maze test, and recognition of new objects, in the NOR test, which may be related to changes in the levels of neurotransmitters.

Intestinal helminths represent a biome capable of modulating the immune system in very different ways. While *H. dimunuta* is considered as a low pathogenic or non-pathogenic parasite, strong host immune responses are activated to prevent parasite establishment during invasion with haematophagic nematodes [20]. As a result, host immunity focuses on eliminating the parasite or confining it to minimize host damage. In addition, our results may differ from previous findings as the ability of rodents to learn a spatial or a discrimination task may also dependent on the type of parasite, the stage and level of infection, species of animal model and experimental protocol.

In the present study, the *H. diminuta* infected rats demonstrated lower striatal levels of serotonin than the control rats, which may be responsible for changes in animal behavior. Serotonin is a key neurotransmitter in the CNS and an important regulatory hormone controlling a broad range of physiological functions. 5-HT affects mood, sleep and anxiety, and peripherally, is involved in the modulation of gastrointestinal motility [21–23]. The enteric serotonergic system plays a principal role in the host-parasite interaction, impacting immunity and thereby the outcome of infection. It is believed that 5-HT is important for intestinal function, and approximately 90% of serotonin is formed in enterochromaffin cells in the gut in humans. Martin et al. [24] report that the central and peripheral pools of serotonin are anatomically separated because 5-HT cannot readily cross the blood-brain barrier.

When infected with helminth parasites, the mammalian gut triggers immune responses to eradicate the parasite, reduce tissue damage, and restore gut homeostasis. These are accompanied by discrete changes in the enteroendocrine and nervous systems, especially those concerning serotonergic signaling. Abdel Ghafar et al. [25] report a significant decrease in serotonin, noradrenaline and dopamine content in the brain in mice experimentally infected with *Schistosoma mansoni*, *Toxocara canis* or *Trichinella spiralis*.

Parasite-induced changes in brain monoamine levels e.g. brain serotonin signaling, may be associated with changes in host behavior. This adaptive mechanism probably increases parasite transmission to the predatory final hosts. As indicated in the literature, these changes may develop in two opposite directions, as parasite invasions have been associated with both increased and decreased serotonergic activity [26–29]. Even so, invasions are known to influence the serotonergic signaling in the brain of the host, thus altering its behavior; indeed, Helland-Riise and co-workers [26] found that infection with brain-encysting trematodes of the species *Euhaplorchis californiensis* caused a decrease in raphe serotonergic neurotransmission in the intermediate host: the California killifish.

*Euhaplorchis californiensis* infection has also been observed to influence serotonin level by Mettrick and Cho [27], who report that higher 5-HT levels in the parasitized gut and blood than in uninfected animals. Moreover, serotonin is important in controlling worm behavior and metabolism; it influences the migration of the worm by regulating of neuromuscular activity and the carbohydrate metabolism of its helminth.

Our data show that infestation by *H. diminuta* not only affects the serotonergic system but also influences noradrenergic neurotransmission. The noradrenergic system regulates various behavioral functions, including selective attention, memory storage and retrieval, vigilance and mood [30]. Noradrenergic neurons in the CNS originate mainly from the locus coeruleus and project to different brain areas, including the cortex, hippocampus, amygdala, thalamus, and hypothalamus. Deregulation of noradrenergic transmission in the brain may result in dementia or Alzheimer's Disease. One emerging idea proposes that Alzheimer's Disease is preceded by abnormal hyperactivation of the LC, resulting in over-secretion of NA in the cortex [31].

In the present experiment, *H. diminuta* invasion was associated with lower levels of NA in the prefrontal cortex and the cerebellum than in infected animals, thus influencing sympathetic nerve function and NA secretion. Similarly, other studies have found that intestinal

inflammation in response to parasitic infection altered noradrenaline release. In the myenteric plexus of *Trichinella spiralis*-infected rats, a significant suppression of NA release was noted on the sixth day post infection, which persisted until 100 days post infection [32].

The impact of *H. diminuta* on the noradrenergic system may indicate its potential in studies of the role of parasitic infections on neurodegenerative diseases and suggests that it may offer some protective potential in this regard. Experimental studies performed by Williamson et al. [15,17] suggest that infection may protect rats against memory loss, while Pillet et al. [33] note that abnormal noradrenergic neuron structure and function are closely related to the etio-pathology of Alzheimer's Disease.

Cytokines may induce depression by affecting serotonergic and noradrenergic system. Increased noradrenergic activity and decreased serotonin level in the brain, caused by pro-inflammatory cytokines, are the typical changes seen in patients with depression [34,35]. The significant decrease in noradrenaline and serotonin observed in our present experiment may explain the motor, behavioral and emotional changes observed in the rats infected with *H. diminuta*. Infection with this low pathogenic tapeworm appears to positively influence spatial memory and new object recognition; in addition, the infected animals develop a greater level of anxiety and move more carefully and slowly.

Behavioral changes in infected rats may be a result of alteration in cytokine transcripts and in consequence of changes in the neurotransmitters level. Regarding the expression of genes, stable negative trends connected with parasitic infection are evident for mRNA encoding cyto-kines in hippocampus, but only for IL-6, traditionally described as a proinflammatory cytokine the reduction in the expression were statistically significant. This may not only occur through the parasite exerting a direct effect on the host organism, but probably also due to changes it may make in the gut microbiome, which interacts with the immune system to affect the CNS. Cytokines may induce opposite biochemical and behavioral effects depending on their concen-tration in specific brain structure. IL-6 is a major cytokine in the CNS secreted by neurons, astrocytes, microglia and endothelial cells as well as important mediator of interaction between the neuroendocrine and immune systems. In vitro IL-6 decreases the differentiation of neural stem cells into neurons [36]. Local brain production of IL-6 is controlled by other cytokines and inflammatory factors as well as by neurotransmitters like noradrenaline and serotonin [37]. Many researchers have emphasized the role of IL-6 in host defense against parasitic infections [38–40]. IL-6 participates in the innate and adaptive immune responses that protect the host from a variety of pathogens. Silver et al. [41] propose that IL-6 is part of a *regulatory loop* that not only initiates inflammation but also limits immunological response in chronic conditions; as such, the lower expression of IL-6 observed in our study may hence be a result of adaptation to chronic infection. The aforementioned research suggests that the parasite exerts an influence on the host, which may lead to changes in the CNS functioning and immunological response. Interactions among brain cells with immune functions such as microglia or astrocytes and peripheral immune cells is necessary to keep gentle balance needed for the proper neurophysio-logical actions. Communication between CNS and the immune system is bidirectional. On the one hand non-neuronal cells response to classical neurotransmitters like glutamate and mono-amines but on the other hand, secretion and responsiveness of neurons and glia to low levels of pro-inflammatory mediators like IL-1 and 6, as well as TNF-α is observed [42].

In our experiment, a decrease in mRNA transcription for IL-6 goes hand in hand with a decrease in serotonin levels. Interactions between IL-6 and brain 5-HT is a complex process which involves opioid peptides, corticotropin-releasing factor and integrated effects of gluta-mate, 3',5'-cyclic AMP, calcium ions, protein kinase C, and other metabolic routs [43]. Pro-inflammatory cytokines affect the 5-HT metabolism by stimulating the enzyme indoleamine 2,3-dioxygenase which leads to a peripheral depletion of tryptophan. Additionally cytokines

can modulate serotonergic signaling by elevating the expression and function of monoamine transporters. Microdialysis study confirm that intraperitoneal injected IL-6 enhanced the 5-HT-like signal in hypothalamus of rats and striatum [43–45].

Behavioral and biochemical data obtained in our study are in line with results presented by Bialuk et al (2018) who revealed the improvement of reference memory in IL-6-deficient mice [46]. Probably endogenous IL-6 may have a physiological role in mechanism responsible for cognitive flexibility acts as a negative regulator of LTP maintenance [47]. Expression of IL-6 in CNS in physiological conditions remains at a low level but elevated levels are reported in several neurodegenerative or psychiatric disease as well as in CNS infection and injury [48].

In our experiment transcripts for IL-1β, IL-10 and TNF-α were present in similar albeit slightly lower amounts in most of the examined CNS structures in infected animals compared to parasite-free rats. However, to better determine the host immune response and the true effect of parasitic infection on cytokine levels, future studies should examine their concentration in the blood and CNS tissues by appropriate analytical methods.

Our results showed that infection with the tapeworm *H. diminuta* influences spatial memory and new object recognition. At the same time, the infected animals develop a greater level of anxiety and move more slowly. Visible changes in the exploratory and motor behavior and cognitive abilities of infected animals may be related to the decreased noradrenergic and serotonergic neurotransmission in the brain structures; however, although the precise nature of this process remains unclear, it may be associated with the modification of the intestinal microbiome. The research we presented has several limitations which make it difficult to unambiguously interpret the results. Inter alia, linking information about interactions with the host microbiome are omitted as well as factors affecting gut neurotransmitters metabolism. Another limitation is lack of research assessing the neurotransmitter metabolome in the gut of infected vs uninfected rats as well as absent information on behavioral assessment the same animal before and during early and late infection.

The interaction between the microbiome and tapeworms remains poorly characterized. It was indicated that colonization of the rats with *H. diminuta* causes a shift in the microbial community, primarily characterized by changes in the relative contributions from species within the *Firmicutes phylum*. Specifically, colonization with the helminth is associated with increased Clostridia and decreased Bacilli [4]. In the same study authors designed the experimental to evaluate the effects of *H. diminuta* on the rat's microbiome following exposure to a mild inflammatory challenge induced by LPS. The results of both experiments support the assertion that colonization with a rat tapeworm influences the microbiome of the host, resulting in shifts in community structure that affects 20% of the total organisms present. On the other hand Aivelo and Norberg (2018) observed that *H. diminuta* had negative associations with several bacterial orders, whereas closely related species *H. nana* had positive associations with several bacterial orders [49]. These study show that Hymenolepis tapeworms were associated with differences in microbiome composition. The presence of *H. diminuta*—but not the closely related species *H. nana*—is correlated with a markedly different microbiota community in analyzed animals. These studies prove the influence of *H. diminuta* on the structure of the intestinal microbiome in infected animals.

Williamson et al. (2016) began pioneering research on the potential interaction between tapeworm infection, the microbiome, and the neurobiology and behavior of infected rats [15]. This was the first experimental evidence illustrating that helminths modulate neuroimmune responses within the brain, with significant consequences for behavior. They noted that helminths prevent exaggerated peripheral and central innate immune responses and alter the gut microbiome, which most probably interacts with the immune system to affect the CNS. It was shown that colonization of pregnant rats with *H. diminuta* attenuated the exaggerated brain

cytokine response of the offspring to bacterial infection, and that combined with post-weaning colonization of offspring with helminths prevented enduring microglial sensitization and cognitive dysfunction in adulthood. Simultaneously, helminths had no overt impact on adaptive immune cell subsets, whereas exaggerated innate inflammatory responses in splenic macrophages were prevented. They also observed evident increase in gene expression of anti-inflammatory IL-4 mRNA along with the macrophage marker CD163, which is associated with inflammation resolution only in neonatally-infected rats with helminths. Provided by Willamson et al. (2016) data confirm that *H. diminuta* do not compromise immune function and most probably inhibit hypersensitive immune responses. Moreover, tapeworms altered the effect of neonatal infection on the gut microbiome. These study indicates potential mechanisms by which tapeworms might exert therapeutic benefits in the treatment of neuroinflammatory and cognitive disorders. Similarly to aforementioned authors in our study we observed no significant changes in the body weight or in mortality by infection in infected rats. This confirms a lack of adverse side effects by these tapeworms. Taking into account most recent evidences of connections between the gut microbiota and the brain involving multiple biological systems, and possible contributions by the gut microbiota to neurological disorders [50] we can speculate that helminths, which are able to change the microbiome structure of the host and impact the host neurotransmission and behavior. However, the exact mechanism of this action still remains unsolved. We can only speculate that changes in the microbiome composition of the *H. diminuta* infected rats may be associated with observed by us changes in the level of neurotransmitters and behavior.

In addition, recent studies [51] indicates the link between host immunity, the gut bacteria and helminth-evoked suppression of colitis in mice model. Authors speculate that reduced efficacy of helminth therapy could be enhanced by combination with a probiotic matched to compensate for dysbiosis in a particular individual.

Intestinal parasites might manipulate host behavior by improvement their own dissemination and/or by influencing gut microbiome and host immune system which in turn influence the CNS functioning. We believe that our results provide some new data on the complex host-parasite crosstalk; this may serve as the basis and show directions for further research on the influence of parasites on brain function and the development of diseases in which a key role may be played by noradrenergic system dysfunctions. Our research clearly indicates that the intestinal macrobiome can influence the behavior of the host. Therefore, in the near future it may turn out that the role of the macrobiome in the functioning of the nervous system may be just as significant as that of the microbiome.

## 4. Materials and methods

### 4.1. Ethics statement

All procedures were carried out in compliance with the directives of the National Research Council (NRC) Committee for the Update of the Guide for the Care and Use of Laboratory Animals (8th edition; National Academies Press, 2011) and approved by the Ethical Committee for Animal Experiments at Medical University of Warsaw. The ARRIVE guidelines 2.0 (Animal Research: Reporting of In Vivo Experiments) were implemented to ensure the highest quality of research standards.

### 4.2. Animals and treatment

Male Lewis rats, aged approximately three months, were used as definitive hosts for adult *H. diminuta*. All rats were kept in plastic cages in the Medical University of Warsaw laboratory animal facilities and were provided feed and water *ad libitum*.

Briefly, six-week-old *H. diminuta* cysticercoids were extracted from dissected *Tenebrio molitor* beetles under a microscope (magnification 100 x) and used to infect three-month-old rats by oral uptake of six cysticercoids of *H. diminuta* suspended in a 0.6% sterile saline per rat. To verify the presence of adult parasites, fecal samples were taken from the animals and examined for the presence of tapeworm eggs under a microscope (magnification 400 x) along the experiment period. The procedure of the animals infection was similar to that described in our previous studies [1–2,5,52–53].

The rats were randomized into two equal groups: one group infected with *H. diminuta* (Hym, n = 7) and an uninfected control group (Con, n = 8). Both groups of rats were housed individually in plastic breeding cages in an air-conditioned room at 24˚C with 12 h dark-light cycle. Both infected animals and those from the control group received standard food (Labofeed, Kcynia, Poland) and tap water *ad libitum*. Parasite infestation was periodically monitored by fecal pellet examination using the fecal flotation test.

At 18 months old, the adult Lewis male rats were used for behavioral testing, after which the rats were sacrificed by decapitation. The body weights were recorded before sacrifice. CNS tissues: hippocampus, prefrontal cortex, striatum, hypothalamus, cerebellum and medulla oblongata were immediately dissected, weighed, quickly frozen on dry ice and stored at -80˚C for future analysis.

## 4.3. Behavioral tests

A detailed experimental diagram is shown in Fig 7. All behavioral test was video tracked using Noldus EthoVision XT10 system (detection settings: center-point detection, dynamic subtraction, sample rate: 8.33 per sec).

**4.3.1. Open Field (OF) test.** The Open Field (OF) test originally developed by Hall [54] allows quantification of animal exploration, as well as various locomotor parameters, such as

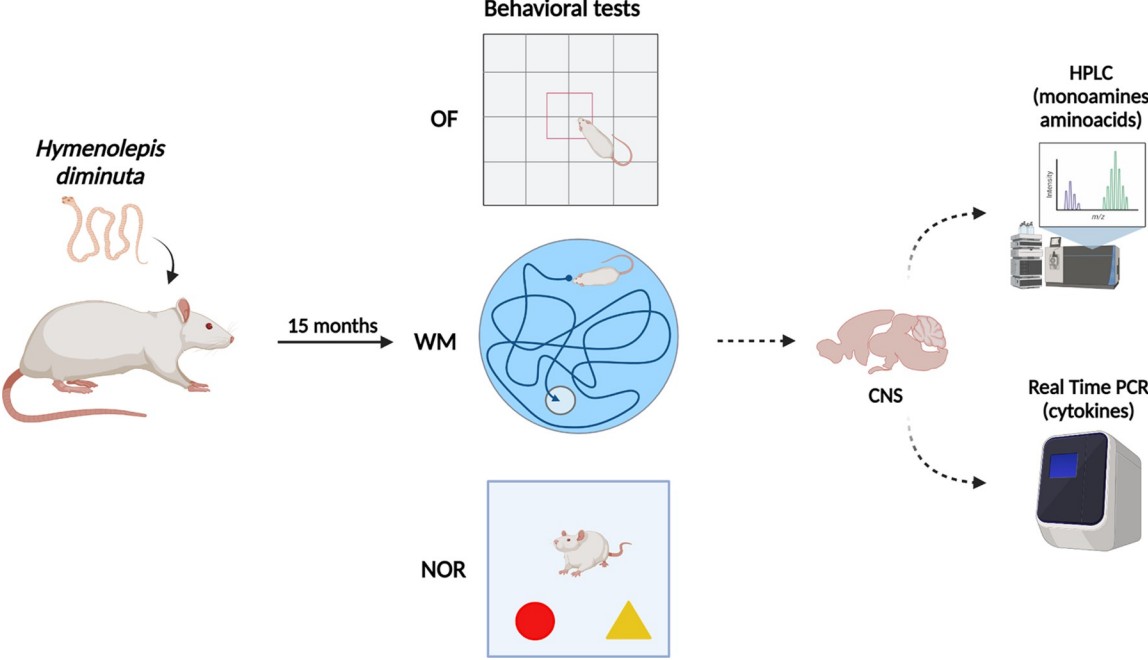

**Fig 7. The graphical course of the experiment (created with BioRender.com).** 15 months after the tapeworm infection, the animals were subjected to behavioral tests, after which the biochemical changes in the structures of the CNS were analyzed using HPLC and RT-PCR.

total distance traveled or time spent on moving. OF also allowed the animals to become habituated to the apparatus, which was subsequently used during the Novel Object Recognition (NOR) test performed the following day. This shortened the time needed for the experiments.

For the OF test, a rat was placed in the center of a gray open field bounded by walls measuring 100 cm x 100 cm x 40 cm (length x width x high). The test apparatus was located in an acoustically-isolated experimental room illuminated by diffused light. On the test day, each rat was placed at the center of arena; the locomotor activity of the exploring animal was monitored for eight minutes with a video camera located above the OF arena and recorded with the Etho-Vision XT10 software (Noldus). The behavior of the animal e.g. frequency of grooming and climbing during all session was scored by observers blinded to the experimental plan. After the end of the 8-minute session, defecation and urination was assessed by experimenter. Rats that spent less time in central zone were considered to display anxiety-like behavior. In order to eliminate odor traces, the field had been cleaned with 70% alcohol before each animal was tested.

**4.3.2. Novel Object Recognition (NOR) test.** The Novel Object Recognition (NOR) test first described by Ennaceur and Delacour [55] with some modifications was performed is used to assess the learning and memory processes in animals and helps in the assessment of neuro-psychological changes. The study was carried out similar to the procedure previously described [56] in the apparatus used for the OF test; however, the it was modified with the addition of two objects for exploration. After each trial, the objects used in the experiment and the testing box itself were disinfected with a 70% solution of ethanol. The test consisted of three stages: habituation, in this case fulfilled by the OF test, familiarization and choice phases. The study was conducted individually for each animal. During the familiarization phase, the animals were placed against the wall opposite another wall where two identical objects (towers made of Lego blocks) marked as A1 and A2 were placed (located at opposite corners of the apparatus). The animals were allowed to freely explore the apparatus for three minutes. On the next day, one of the objects was replaced with another unknown object (object B—glass bottle). New and familiar items were presented in different colors, structures and shapes. Object exploration was classified as the rat placing its head at a distance of 2 cm from any object. During the experiment, frequency and the time spent exploring individual objects during each phases and the total time spent exploring both objects were measured, and the Discrimination Index was calculated for the choice phase as the difference between the time (t) taken to explore the novel and known objects [DI = (tB-tA1)/(tB+tA1)]. The Index of Global Habituation [GHI = (tA1 +tA2)/(tB+tA1)], which compares the time spent studying two subjects during the familiarization phase with the time spent during the choice phase, was also estimated. The Recognition Index, the time spent on researching the new object in relation to the total exploration of the objects [RI = tB/(tB+tA1)], was also determined.

**4.3.3. Water Maze (WM) test.** A modified version of the Morris Water Maze (WM) test was used to evaluate the influence of *H. diminuta* infestation on learning and memory in rats [57,58]. To reduce the stress and natural aversion of animals to the water, the rats were habituated to the experimental conditions 24 hours before the behavioral tests. The WM test was performed in a circular pool (150 cm in diameter; 50 cm high) filled with water (30 cm height) maintained at 23°C and divided conventionally into four quadrants (Southeast–SE, Southwest–SW, Northeast–NE, Northwest–NW). The maze was located in a room and was surrounded by several objects as spatial coordinates e.g., shelves, screens and posters.

During the water acquisition phase, the rats were trained for four consecutive days (four trials per day) to locate a hidden transparent round, plexiglass platform (10 cm diameter) submerged 1 cm below the water surface in the middle of the SE sector. In each trial, the rat was randomly placed into the pool facing the wall at an equidistant cardinal points in a

pseudorandom schedule. If the rat found the platform, it could stay there for 15 seconds to sufficiently assess distal cues. When the animal failed to find the platform within 60 seconds, the experimenter manually guided the rat to the platform for 15 s. Then each rat was removed from the tank, dried by wiping with a cloth, and returned to its holding cage.

During the spatial memory test assessed on the fifth day (trial 17, Memory Test I) the platform was removed from the pool, and the animals searched for the absent platform for 60 seconds.

Two days later (day 8, trials 18–21), a Remainder Test was conducted. The platform was placed again in the SE sector, and the rats recalled the location of the underwater goal.

A reversal trial was carried out on day 9 (trials 22–25). The platform was transferred to the center of the NW sector and the animals learned to find its new position. On day 10 (trial 26, Memory Test II) the platform was removed again, and the rats swam only once for 60 seconds, searching for the absent platform. The time spent in the target quadrants and crossings under earlier position of platforms in quadrants SE and NW were used as an indicator of memory retention.

Sensorimotor skills and motivation were assessed on day 11 during a cued task (visible platform task). In this task, the daily training session consisted of four attempts. The visible platform was located 1 cm above the water level and was moved successively between four sectors. Three randomly-assigned starting points were located at the perimeter of the pool, in the center points of the sectors without the platform. The test was ended when the rat reached the platform, or after 60 seconds.

Latencies to find the platform, a number of visits in a target area and the time spent in the goal quadrant, swim paths and velocity were recorded using a video camera and EthoVision XT10 software (Noldus).

## 4.4. HPLC

HPLC was used to estimate the concentrations of monoamines and amino acids in selected rat CNS tissues: hippocampus, prefrontal cortex, striatum, hypothalamus, cerebellum and medulla oblongata.

**4.4.1. Sample preparation.** Before analysis, brain samples were homogenized using ultrasonic cell disrupter (VirSonic 60; VirTis, USA) in a mixture (1000 μl) containing ice-cold 0.1 N perchloric acid (HClO4) and 0.05 mM ascorbic acid, and then centrifuged to precipitate proteins (Heraeus Labofuge 400 R, Heraeus Instruments, Germany; centrifugation conditions: speed of 13,000 x g, time of 15 min, temperature of 4˚C). After filtration using syringe membrane filters with pore size 0.2 μm (Puradisc; Whatman, UK), the supernatant was collected and used to perform biochemical analyses.

**4.4.2. Assay of monoamine concentrations.** To determine the concentration of serotonin (5-HT) and its metabolite 5-hydroxyindoleacetic acid (5-HIAA), dopamine (DA) and its metabolites 3,4-dihydroxyphenylacetic acid (DOPAC) and homovanillic acid (HVA), and noradrenaline (NA) and its metabolite 3-methoxy-4-hydroxyphenylglycol (MHPG) a 20 μl aliquot of supernatant was injected into the HPLC apparatus with electrochemical detection (HPLC-ED) system.

The HPLC system consisted of a delivery pump (Mini-Star K-500; Knauer, Berlin Germany), an autosampler automatic injector (LaChrom L-7250; Merck-Hitachi, Darmstadt/ Tokyo, Germany/Japan), an electrochemical detector (L-3500A; Merck-Recipe, Darmstadt/ Munich, Germany) set at a potential of +0.8 V *vs* an Ag/AgCl reference electrode. The mobile phase comprised 32 mM sodium phosphate buffer (Sigma-Aldrich, St. Louis, MO, USA), 34 mM citric acid buffer (Sigma-Aldrich, USA), 1 mM octane sulfonic acid buffer (Sigma-

Aldrich, USA), 54 μM ethylenediaminetetraacetic acid (EDTA) buffer (Sigma-Aldrich, USA) in ultrapure water (18 MΩ·cm) containing 12% methanol solution (Merck, Germany). Mono-amines were separated using EC 250/4 Nucleosil 100–5 C18AB (250 mm length x 4 mm internal diameter, 5 μm particle size, 100Å) HPLC analytical column (Macherey-Nagel, Germany) and mobile phase flow rate maintained at 0.8 ml/min. Chromatograms were recorded and integrated by use of the computerized data acquisition Clarity software (version 5.0; DataApex, Prague, Czech Republic). Samples were quantified by comparison with standard solutions (external calibration). All used monoamines standards were purchased from Sigma-Aldrich, USA. The final amount of monoamines/metabolites in the tissue sample was expressed as pg/mg wet tissue.

**4.4.3. Assay of Amino acid concentrations.** High Performance Liquid Chromatography with electrochemical detection (HPLC-ED) was used to determine the concentration of the following amino acids: glutamic acid (GLU), aspartic acid (ASP) gamma-aminobutyric acid (GABA), alanine (ALA), histidine (HIS), serine (SER), arginine (ARG), threonine (THR), glutamine (GLN) and taurine (TAU) in the medium. The chromatograph system consisted of an electrochemical detector with glassy-carbon working electrode (EC 3000, Merck), the auto-sampler (Primade 1210, Hitachi) and the pump (Primade 1110, Hitachi). The mobile phase comprised 45 mM disodium phosphate (Sigma–Aldrich) and 0.15 mM ethylenediaminetetraacetic acid (EDTA; Sigma–Aldrich) with 24% methanol (Merck). To obtain agents for derivatisation o-phthaldialdehyde (OPA, 22 mg, Fluka) was diluted in 0.5 ml of 1 M sodium sulphite, 0.5 l of methanol, 0.9 ml of 0.1 M sodium tetraborate buffer (adjusted to pH 10.4 using 5 M sodium hydroxide). The preparation of the mobile phase and the derivatising agents were based on the method developed by Rowley et al. [59] with minor modifications. Prior to application onto the column, the derivatising agent (20 μl) was reacted with amino acid standard and with supernatant samples for 15 min at room temperature. Separation of amino acids was achieved with a reverse phase column (Luna; 5 μm C18(2) 100A; 250 mm length × 4 mm internal diameter; Phenomenex) and mobile phase flow rate maintained at 0.8 ml/min. Samples were quantified by comparison with standard solutions (external calibration) and concentrations calculated with Primade (Merck). All standards were purchased from Sigma–Aldrich. The final amount of amino acids in the tissue sample was expressed as ng/mg wet tissue.

## 4.5. Real-time polymerase chain reaction (Real-Time PCR)

The mRNA expression of pro and anti-inflammatory cytokines TNF-α, IL-1β, IL-6 and IL-10 were assessed using real-time PCR. Total RNA was extracted from the tissue using the TRI Reagent (Sigma-Aldrich, USA), in accordance with the manufacturer's instructions. The concentration of isolated RNA was determined spectrophotometrically at 260 nm. Single-strand cDNA was synthesized from total RNA using PrimeScript RT reagent Kit (Takara Bio Inc, Japan) in a Labcycler (SensoQuest GmbH, Germany). The incubation conditions for reverse transcription were 15 min at 37°C followed by 5 s at 85°C. Following the reverse transcription reaction, cDNA products were stored at –20°C until use.

Real-time PCR was conducted using a Rotor Gene Q 5plex HRM System (QIAGEN Research, Inc., Netherlands). The cDNA was amplified with gene-specific primers designed using Primer BLAST software (NCBI, USA) (Table 4). The housekeeping gene (glyceraldehyde 3-phosphate dehydrogenase, GAPDH) was used to normalize gene expression levels.

The PCR mixtures contained 1 μl of previously reverse-transcribed cDNA along with 10 μl of FastStart Essential DNA Green Master (Roche Molecular Systems, Inc., USA) and 1.25 μl (10 μM) of each primer in a total reaction volume of 20 μl. The amplification protocol was as follows: initial denaturation at 95°C for 10 min; 40 cycles at 95°C for 15 s, 58°C for 15 s, and

**Table 4. Sequences of gene specific primers used in Real Time PCR (F-forward, R-reverse).**

| Target gene | Sequences of gene-specyfic primers (5'>3') | Predicted product lenght (bp) |
|---|---|---|
| GAPDH | F GCTCTCTGCTCCTCCCTGTTCT<br>R TCCGATACGGCCAAATCCGTT | 111 |
| IL-6 | F GCCAGTTGCCTTCTTGGGAC<br>R CCTCTGTGAAGTCTCCTCTCCG | 83 |
| IL-10 | F CCATTCCATCCGGGGTGACA<br>R TCAGCTCTCGGAGCATGTGG | 71 |
| IL-1β | F ACTCGTGGGGATGATGACGACC<br>R GGTCAGACAGCACGAGGCAT | 102 |
| TNF-α | F GTGGCTCTGGGTCCAACTCC<br>R CCGCAATCCAGGCCACTACT | 91 |

72˚C for 15 s. The reaction melting-curve analysis was applied to all reactions to ensure the consistency and specificity of the amplified product. All amplifications were carried out in duplicate. The relative expression of the genes was determined according to Pfaffl method[60].

## 4.6. Statistical analysis

The results were analyzed using TIBCO Statistica 13.3. Group differences were assessed by analysis of variance (ANOVA). The comparisons between groups were performed using post-hoc Newman–Keuls (NK). Repeated measures two way ANOVA was used to analyze repeated observations. Neurotransmitter concentration and mRNA expression for the target genes were analyzed with the Mann-Whitney U-test, comparing control and infected groups. Correlations between the measures of cognitive function and neurotransmitters protein levels and cytokines expression were analyzed by Pearson's correlation. All the data are presented as mean ± SEM. The results were considered statistically significant at $p < 0.05$.

## Author Contributions

**Conceptualization:** Kamilla Blecharz-Klin, Ilona Joniec-Maciejak, Daniel Młocicki, Dagmara Mirowska-Guzel.

**Data curation:** Kamilla Blecharz-Klin, Ilona Joniec-Maciejak, Dagmara Mirowska-Guzel.

**Formal analysis:** Kamilla Blecharz-Klin, Agnieszka Piechal, Adriana Wawer, Justyna Pyrza-nowska, Dagmara Mirowska-Guzel.

**Funding acquisition:** Daniel Młocicki, Dagmara Mirowska-Guzel.

**Investigation:** Kamilla Blecharz-Klin, Magdalena Świerczyńska, Agnieszka Piechal, Adriana Wawer, Ilona Joniec-Maciejak, Justyna Pyrzanowska, Ewa Wojnar.

**Methodology:** Kamilla Blecharz-Klin, Magdalena Świerczyńska, Agnieszka Piechal, Adriana Wawer, Ilona Joniec-Maciejak, Justyna Pyrzanowska, Ewa Wojnar, Anna Sulima-Celińska, Dagmara Mirowska-Guzel.

**Project administration:** Kamilla Blecharz-Klin, Daniel Młocicki, Dagmara Mirowska-Guzel.

**Resources:** Daniel Młocicki, Dagmara Mirowska-Guzel.

**Software:** Dagmara Mirowska-Guzel.

**Supervision:** Daniel Młocicki, Dagmara Mirowska-Guzel.

**Validation:** Kamilla Blecharz-Klin, Magdalena Świerczyńska, Dagmara Mirowska-Guzel.

**Visualization:** Anna Zawistowska-Deniziak.

**Writing – original draft:** Kamilla Blecharz-Klin, Dagmara Mirowska-Guzel.

**Writing – review & editing:** Anna Zawistowska-Deniziak, Daniel Młocicki, Dagmara Mirowska-Guzel.

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
