## [Decision Letter · Decision Letter 0]

23 Nov 2021

Dear Dr. Mlocicki,

Thank you very much for submitting your manuscript "Infection with intestinal helminth (Hymenolepis diminuta) impacts exploratory behavior and cognitive processes in rats by decreasing noradrenergic and serotonergic neurotransmission." for consideration at PLOS Pathogens. As with all papers reviewed by the journal, your manuscript was reviewed by members of the editorial board and by several independent reviewers. In light of the reviews (below this email), we would like to invite the resubmission of a significantly-revised version that takes into account the reviewers' comments.

There were numerous issues raised with rationale and completeness of the experiments

We cannot make any decision about publication until we have seen the revised manuscript and your response to the reviewers' comments. Your revised manuscript is also likely to be sent to reviewers for further evaluation.

Sincerely,

De'Broski R Herbert

Guest Editor

PLOS Pathogens

P'ng Loke

Section Editor

PLOS Pathogens

Kasturi Haldar

Editor-in-Chief

PLOS Pathogens

orcid.org/0000-0001-5065-158X

Michael Malim

Editor-in-Chief

PLOS Pathogens

orcid.org/0000-0002-7699-2064

There were numerous issues raised with rationale and completeness of the experiments

Reviewer's Responses to Questions

**Part I - Summary**

Reviewer #1: The authors evaluated the effects of intestinal tapeworm infection on behavior and central nervous system function in rats. Various aspects of learning and memory were assessed by measuring performance on several behavioral tests. The quality and quantity of overall activity in rats was measured in the exploratory behavior test.

Changes in brain activity induced by tapeworm infection were monitored by levels of neurotransmitters, their metabolites and amino acids in different parts of the brain. Gene expression of pro- and anti-inflammatory cytokines in brain neural tissue was also measured.

Using behavioral neuroscience tools, the authors showed that the parasite significantly altered exploratory behavior, decreased speed and total distance of movement, decreased frequency in the central zone, and thus increased anxiety in rats.

At the same time, better spatial memory and recognition of new objects were found. Reduced noradrenaline and serotonin levels were also identified, as well as low IL-6 expression in the hippocampus.

These neuro-immunological-parasitological data provide new perspectives for further studies on the noradrenergic system under intestinal parasite infection.

The study was performed at very good level, however presentation need to be improved.

The strength of the paper

The subject is novel and nowadays need to be explored, especially as parasitic infection or parasitic derived compounds were noticed for their immune-regulatory activity for neurodegenerative and autoimmune disorders of neural tissue. Therefore the data of behavioral changes induced by active parasitic infection may raise interest not only in scientific communities but also in the open audiences. The data could also be valid for explanations of the host-parasite relationship at the evolutionary level and their molecular adjustment. Data concerning behavioral results are presented and discussed sufficiently to support conclusions.

The methodology is appropriate and well described. In general, the aim of the study is provided.

Weaknesses of the presentation

Based on the methodology used in this research, more specific questions would be worth asking. The relationship between neurotransmitter levels and cytokine production in the brain needs to be presented in the introduction. It is not explained why different parts of the brain were assessed for neurotransmitter levels. Some question are raised.

1. Is there any relationship with rat behavior and the distribution of serotonin or noradrenaline and IL-6?

2. Why were the major metabolites of noradrenaline measured? No link has been suggested between noradrenaline and immune response mechanisms, including local cytokine release in the brain.

3. How to link low levels of serotonin, noradrenaline, and IL-6 with increasing levels of anxiety in parallel with improved spatial memory and better recognition of new objects.

4. What is the intestinal parasite function for the downregulation of neurotransmitters and cytokine production?

A weakness of the manuscript is description of results. Figures have a very poor and short legend. The text lacked proper reference to the parts of the graph that were analysed.

Reviewer #2: Overall, the concept behind the study is interesting and in need of examination. However, the data are largely buried in the text rather than being cogently summarized graphically and doesn’t match the statistical tests performed and reported in the text. The rationale and meaning of the behavioral tests and HPLC data is not clearly laid out. This needs to be addressed with significant editing, including additional graphs or tables, to make it clear how the infected and uninfected animals’ behavior changed and how the neurotransmitter content might relate to that. The manuscript is a little muddled between the two possibilities that (1) parasites directly manipulate host behavior to benefit their dissemination (which was not addressed by this study – but could be with other behavioral assays that assess social and feeding behavior against parasite burden and dissemination) and (2) the parasites, through interactions with the host microbiome and immune system change brain chemistry and function to affect the mood, cognitive function and behavior of the host (which was somewhat evaluated by this study, but missing the causative components between the parasite and the behavioral/brain changes – it could be microbiome changes affecting gut neurotransmitter metabolism that could affect the CNS, it could be changes in peripheral inflammation affecting the CNS). Linking information is missing. The assertion that infected animals have greater anxiety needs to be bolstered by additional data, as time in the center of the open field alone is controversial as an indicator of anxiety. The authors need to provide some data regarding the health status of the infected animals overtime more explicitly because malaise can appear similar to anxiety (Body weights and/or body condition score, locomotor data collected at different times shown graphically). While their links to neurodegenerative disorders are intriguing it seems premature based on the current data. And if tapeworm does increase anxiety, would be counterproductive in neurodegenerative disorders often comorbid with depression and anxiety.

**Part II – Major Issues: Key Experiments Required for Acceptance**

Reviewer #1: The detailed hypothesis of this research should better refer to the measured parameters.

The molecular results were not fully analyzed and concluded.

Reviewer #2: 1. Use of open field alone to determine anxiety is insufficient, since malaise due to infection could explain slow velocity and reduced center time. Additional tests such as the elevated plus maze and light dark assay would strengthen this argument. Evaluating stress hormones such as corticosterone in infected vs uninfected rats both outside of testing, and following a stressful test could also help strengthen the argument. If you are not able to do such experiments, then the discussion needs to acknowledge this. Body wieghts would also help, as would a lack of difference in the rotarod test, or simply more clearly laying out all of the motility data in your study collected at different times graphically.

2. The lack of consideration for the timecourse is a drawback of the study's design, where rats were infected for 15 months. For behavior it is possible to assess the same animal before and during early and late infection. If this cannot be done, then this limitation should be more clearly acknowledged in the discussion. The authors should consider collaborating with a lab equipped to do microdialysis measurements in the brain - this could provide a timecourse of changes in brain metabolites without the need to sacrifice animals at defined timepoints. It may also provide a more nuanced assessment of changes in specific brain nuclei. A more quantitative assessment of the parasite burden may also help here.

3. The manuscript is lacking the causative mechanism for brain changes with infection. Assessing the neurotransmitter metabolome in the gut of infected vs uninfected, perhaps also the microbiota - specific bacterial species affect these processes, might be informative towards linking the infection with changes in brain neurotransmitter metabolites.

**Part III – Minor Issues: Editorial and Data Presentation Modifications**

Reviewer #1: Please better present results included in Table 3; however most of them are not statistically significant, but stable negative trends are evident for mostly measured cytokines.

The discussion is conducted too wide and several information is out of the paper scope or may be presented in introduction as general knowledge (lines 267-275; 325-331; 335-337; 350-358). The narration seems to be typical for a review paper and is not focused on obtained results.

Please consider citation of the following papers.

Kong, E., Sucic, S., Monje, F. et al. STAT3 controls IL6-dependent regulation of serotonin transporter function and depression-like behavior. Sci Rep 5, 9009 (2015). https://doi.org/10.1038/srep09009

Roohi, E., Jaafari, N. & Hashemian, F. On inflammatory hypothesis of depression: what is the role of IL-6 in the middle of the chaos?. J Neuroinflammation 18, 45 (2021). https://doi.org/10.1186/s12974-021-02100-7

Gruol DL. IL-6 regulation of synaptic function in the CNS. Neuropharmacology. 2015;96(Pt A):42-54. doi:10.1016/j.neuropharm.2014.10.023

Table 2 may be presented as Supplementary data.

The discussion is conducted too wide and several information is out of the paper scope or may be presented in introduction as general knowledge (lines 267-275; 325-331; 335-337; 350-358). The narration seems to be typical for a review paper and is not focused on obtained results.

Several specific comments notes and remarks are suggested in the attached and reviewed version of the manuscript and which are labelled in yellow.

Reviewer #2: 1. Title – Neurotransmission hasn't been directly studied by electrophysiology or pharmamcological approach. I would encourge the authors to reword the title to indicate reduced levels of neurotransmitters.

2. In the abstract and elsewhere in the manuscript, the authors discuss parasites that invade the brain. H. diminuta is not one of those, is it? So how is it altering brain chemistry and behavior? Later in the introduction the authors bring up the idea of the gut-brain axis – this may be a better way to open the abstract.

3. Introduction needs some rewriting. There are two ways that intestinal parasites might manipulate host behavior – to improve their own dissemination, which hasn’t been addressed in this study, or, as the authors indicate, influencing gut microbiome and host immune system in ways that then influence the CNS. The latter is more likely in the case of H. diminuta, but the authors need to provide more specific information about H. diminuta’s life cycle (how does it get into the host) and what is known about how it changes the microbiome and immune system. How might those changes alter immune or brain function?

4. Methods – provide citations for the behavioral assays and infection methodology. Were rats gavaged to deliver the cysticercoids? How do you know 6 are sufficient to induce a lasting infection? Is infection maintained by that initial event or supported by coprophagy in rats?

5. Make Figure 3, which is the entire experimental design, Figure 1 and move NOR test to the middle to indicate testing order (top to bottom).

6. Methods pg. 17 delete the word “attitude”. Change “activities” to “experiments”. The phrase “reduced the discomfort of the animals caused by the additional procedure” should be omitted or reworded – you don’t know if they experience more discomfort or if you alleviated that. Saying it more like “we did it this way to reduce the amount of testing and minimize any stress the animals may experience” is more appropriate.

7. Please indicate in methods when and how grooming, climbing, urination, and defecation were assessed – was this only one time in the open field, or in the home cage? By human observation or tracked in some way by the EthoVison software? What are the units associated with the numbers in parentheses in the results section on pg.6? Please give the unit for time in center also.

8. Give the units for everything in parenthesis in the Results section. Define the meaning of the indicies mentioned for the novel object recognition test. Show that data graphically.

9. In the results, indicate the meaning of each phase of the Morris Water Maze tests. What is each phase assessing? Why did you choose to evaluate each of these aspects of learning and memory? Does H. diminuta affect body temperature (this would affect how uncomfortable being in the water is and therefore how motivating it is to get out)?

10. Please indicate in the methods the EthoVision settings for movement detection.

11. Open Field statistics are not described in the Statistics section of the methods and the statistics generally don’t make sense relative to the way the data is shown throughout the manuscript. F values like ANOVA are indicated in the text in relation to paired data that could be evaluated by a T-Test. If that data is summarized across multiple time points, give T-Test statistics for that, or show graphically all of the timepoint data evaluated statistically in a line graph. If you are evaluating multiple time points within the same group, and intending to compare the two groups over time, it may be more appropriate to do a repeated measures two way ANOVA with between subject effects.

12. Report most of your data graphically or in a table and refrain from reiterating everything in text parentheticals. This manuscript could stand to have additional multi-panel figures or line graphs of the data rather than paired bar graphs.

13. Body weights of rats pre and post infection, ideally throughout the course of infection?

14. Although infection was monitored and infected rats were individually housed, it would be better if the authors could provide a more quantitative assessment of parasite burden through the course of infection and behavioral testing. The rats were infected for 15 months – fluctuations in parasite burden could introduce variability in the data, which should be discussed.

15. In the results, explain your rationale and the importance of each aspect of the Morris Water Maze test that is being evaluated. This data especially could benefit from a more complete graphical presentation. Very often these tests are show across the testing time course, not just the end day or a total.

16. On pg 17 in methods, please list the brain tissue dissected, and how were they frozen, liquid nitrogen or dry ice?

17. The HPLC results – please spell out the metabolite names at first mention, and give the units. These values are the content at death – turnover is arrested at that point. Please explain exactly what you are measuring to determine “MHPG/NA turn over”. Is this simply the ratio of the metabolite vs NA amount measured? What exactly does this indicate? Can you represent this data graphically or in a table rather than in the text.

18. In the discussion you need to discuss the possibility that some of the deficits could be associated with malaise, fatigue/nutritional deficits associated with infection, and may not relate to anxiety or an altered emotionality per se.

19. Discussion page 13 paragraph lines 300-8: Why do you say “especially brain serotonin signaling, may be associated with changes in host behavior”? The next sentence “This adaptive mechanism probably greater parasite transmission to the predatory final hosts.” needs to be rewritten. Do you mean “increases” instead of “greater”? What is the significant of predatory hosts in that statement? The next sentence requires citations where parasites are associated with increased and decreased serotonergic activity.

20. The hippocampus is critical for learning and memory and the Morris Water Maze test is dependent on the hippocampus, while the novel object recognition task is hippocampal -independent. Is there any evidence indicating that altered IL-6 in the hippocampus affects learning and memory? The emphasis in the discussion on IL-6 and anti-parasite immunity actually doesn’t fit here because your parasite is not directly in the hippocampus and could be removed from the discussion.

21. (Discussion/Introduction) What is known regarding how H. diminuta changes the microbiome and immune system, how could those two things could affect brain neurotransmitter/cytokine content in a manner consistent with your data? The authors also need to more clearly define the limitations of their findings in the discussion.

22. Why were males used and not females? Will there be follow up in females?

PLOS authors have the option to publish the peer review history of their article (what does this mean?). If published, this will include your full peer review and any attached files.

Reviewer #1: **Yes: **Maria Doligalska

Reviewer #2: No
---

## [Editor Report · Decision Letter 1]

2 Feb 2022

Dear Dr. Młocicki,

We are pleased to inform you that your manuscript 'Infection with intestinal helminth (Hymenolepis diminuta) impacts exploratory behavior and cognitive processes in rats by changing the central level of neurotransmitters' has been provisionally accepted for publication in PLOS Pathogens.

Best regards,

De'Broski R Herbert

Guest Editor

PLOS Pathogens

P'ng Loke

Section Editor

PLOS Pathogens

Kasturi Haldar

Editor-in-Chief

PLOS Pathogens

orcid.org/0000-0001-5065-158X

Michael Malim

Editor-in-Chief

PLOS Pathogens

orcid.org/0000-0002-7699-2064

The authors have done a nice job of improving this manuscript and addressing the reviewers prior concerns.
---

## [Editor Report · Acceptance letter]

8 Mar 2022

Dear Prof Młocicki,

We are delighted to inform you that your manuscript, "Infection with intestinal helminth (Hymenolepis diminuta) impacts exploratory behavior and cognitive processes in rats by changing the central level of neurotransmitters," has been formally accepted for publication in PLOS Pathogens.

Best regards,

Kasturi Haldar

Editor-in-Chief

PLOS Pathogens

orcid.org/0000-0001-5065-158X

Michael Malim

Editor-in-Chief

PLOS Pathogens

orcid.org/0000-0002-7699-2064